# Exploring Next Generation Probiotics for Metabolic and Microbiota Dysbiosis Linked to Xenobiotic Exposure: Holistic Approach

**DOI:** 10.3390/ijms232112917

**Published:** 2022-10-26

**Authors:** Alfonso Torres-Sánchez, Alicia Ruiz-Rodríguez, Pilar Ortiz, María Alejandra Moreno, Antonis Ampatzoglou, Agnieszka Gruszecka-Kosowska, Mercedes Monteoliva-Sánchez, Margarita Aguilera

**Affiliations:** 1Department of Microbiology, Faculty of Pharmacy, Campus of Cartuja, University of Granada, 18071 Granada, Spain; 2Institute of Nutrition and Food Technology “José Mataix” (INYTA), Centre of Biomedical Research, University of Granada, 18016 Granada, Spain; 3Department of Biochemistry and Molecular Biology II, Faculty of Pharmacy, Campus of Cartuja, University of Granada, 18071 Granada, Spain; 4Department of Environmental Protection, Faculty of Geology, Geophysics, and Environmental Protection, AGH University of Science and Technology, Al. Mickiewicza 30, 30-059 Krakow, Poland; 5Instituto de Investigación Biosanitaria ibs, 18012 Granada, Spain

**Keywords:** Microbiota Disrupting Chemicals (MDCs), microbiota, metabolites, Next Generation Probiotics (NGPs), xenobiotics

## Abstract

Variation of gut microbiota in metabolic diseases seems to be related to dysbiosis induced by exposure to multiple substances called Microbiota Disrupting Chemicals (MDCs), which are present as environmental and dietary contaminants. Some recent studies have focused on elucidating the alterations of gut microbiota taxa and their metabolites as a consequence of xenobiotic exposures to find possible key targets involved in the severity of the host disease triggered. Compilation of data supporting the triad of xenobiotic-microbiota-metabolic diseases would subsequently allow such health misbalances to be prevented or treated by identifying beneficial microbe taxa that could be Next Generation Probiotics (NGPs) with metabolic enzymes for MDC neutralisation and mitigation strategies. In this review, we aim to compile the available information and reports focused on variations of the main gut microbiota taxa in metabolic diseases associated with xenobiotic exposure and related microbial metabolite profiles impacting the host health status. We performed an extensive literature search using SCOPUS, Web of Science, and PubMed databases. The data retrieval and thorough analyses highlight the need for more combined metagenomic and metabolomic studies revealing signatures for xenobiotics and triggered metabolic diseases. Moreover, metabolome and microbiome compositional taxa analyses allow further exploration of how to target beneficial NGP candidates according to their alleged variability abundance and potential therapeutic significance. Furthermore, this holistic approach has identified limitations and the need of future directions to expand and integrate key knowledge to design appropriate clinical and interventional studies with NGPs. Apart from human health, the beneficial microbes and metabolites identified could also be proposed for various applications under One Health, such as probiotics for animals, plants and environmental bioremediation.

## 1. Introduction

Human gut microbiota homeostasis depends on many endogenous and exogenous factors, which induce changes that directly affect microbiota composition, its function and host health and disease states. Some exogenous compounds can interfere with different physiological processes, including hormone signalling pathways, acting as endocrine disruptors. Some of these compounds, known as Microbiota Disrupting Chemicals (MDCs), such as bisphenol A (BPA), are incorporated into the body mainly through the diet (food and materials in contact with food) [1]. The negative impact of dietary xenobiotics on human gut microbiota and the development of physiological disorders are increasingly being studied. Xenobiotics may alter the microbiota through changes in the abundance of the microbial taxa and their released metabolites that lead to a state of dysbiosis, which could be linked to several host disorders, such as metabolic diseases.

Specific prevention and treatments are needed to face this altered microbial pattern and restore homeostasis perturbed by xenobiotic-gut microbiota interactions. Traditional probiotics for clinical interventions have been largely applied as a useful strategy to alleviate adverse effects derived from gut microbiota dysbiosis. However, new high-throughput strategies using the application of NGP have become more relevant in recent years, due to their well-demonstrated restoration effects. In this context, microorganisms such as *Akkermansia muciniphila*, *Faecalibacterium prausnitzii*, *Bacteroides uniformis*, *Bacteroides acidifaciens*, *Bacteroides thetaiotaomicron*, *Clostridium butyricum*, *Prevotella copri*, *Christensenella minuta*, and *Parabacteroides goldsteinii*, have been postulated as NGP candidates, because of their preventive and palliative effects on diseases such as obesity, diabetes, colitis, and liver diseases [2,3,4,5,6,7,8,9,10].

Nowadays, the search for microbial profiles to differentiate dysbiotic and eubiotic states in humans plays a vital role in current clinical microbiology approaches, in terms of prevention and treatment of metabolic diseases. In this context, the analysis and description of trends in microbial populations associated with disease and health states are still open. The interactions between environmental xenobiotics, human gut microbiota, microbial metabolites and host status are highly complex and require holistic approaches to better understand how the gut microbiome and derived metabolites affect host development, health, and disease occurrence.

Xenobiotic transformations induced by different host biological mechanisms may generate a complex metabolic network that affects both the host and the microbiota components. As a consequence, the chemical structures of many of these compounds could be modified, resulting in changes in their bioactivity and half-life in the host organism [11]. Variations in microbial populations, because of dysbiotic states, can affect the metabolism of xenobiotics incorporated from the environment, and metabolites formed during their degradation that could result in other substances potentially more toxic than the original one [12]. Overall, there are still many unknown aspects to elucidate in this field. Although knowledge about the role of the microbiota in the transformation of xenobiotics has increased, there are still many interactions that are not clear in this context. Future challenges lie in identifying these microorganisms, genes and metabolites involved in still unknown metabolic processes [11].

We think that this work is significant and necessary but complex because it tries to integrate different key data available from topics which have been studied independently. There are few data that show the link between the impact of xenobiotics in host health which concern the role of individualised microbiota taxa, pathways, and key metabolites triggering diseases, which are susceptible to being modulated and becoming interventional biomarkers. Therefore, the principal aim of this work was to compile data, and identify and describe the potential association between environmental and dietary xenobiotic exposure, gut microbiota taxa, and gut microbiota metabolites, taking into account implications for host health and approaching novel biological strategies to restore gut microbiota dysbiosis and the dysfunctions induced.

## 2. Results

### 2.1. Extraction of Data and Analysis

After an extensive literature search, links between variations in human gut microbiota taxa and metabolic-endocrine disorders were assessed to understand better the possible relationship underlined by many authors in recent years. In this context, we followed an approach that offered some drivers about how specific changes in the gut microbiota composition could be related to host health.

The analysis of 51 articles involving variation of the main taxa altered in patients suffering metabolic and endocrine-related diseases disclosed 119 different microbial genera (complete data are available in Appendix A).

Certain microorganisms belonging to the genera *Faecalibacterium*, *Bifidobacterium*, *Bacteroides*, *Roseburia*, *Alistipes*, and *Akkermansia* showed an upward trend in those cases in which individuals were not affected by metabolic pathologies. However, other microbial genera such as *Lactobacillus*, *Escherichia*, *Blautia*, *Streptococcus*, and *Klebsiella* showed an upward trend in individuals affected by the metabolic-endocrine disorders studied here (Figure 1).

### 2.2. Combined Analysis of Microbiota Taxa and Metabolites in Metabolic Diseases

The gut microbiota could play a central role in the host physiology by producing specific metabolites and/or modulating host metabolism. Perturbations of gut microbiota taxa seem to contribute to alterations in several host metabolic pathways and subsequently to the development and severity of certain common pathologies. In this context, the analysis and data extraction performed allowed us to compile the main variations of predominant microbiota taxa and key metabolites information available from patients suffering metabolic-related diseases, such as obesity, diabetes, cardiovascular disorders, NAFLD, NASH and inflammatory gut diseases. Only seven studies contained combined data on microbiota and metabolites.

Multiple authors have successfully described changes in microbial taxa associated with metabolic-endocrine disorders. The thorough analysis of recent studies focused on the composition of the gut microbiota and resulting metabolites related to the biological and physiological impact on host health status also showed correlations that indicated links between these factors.

Figure 2 and the extracted analysis of the studies selected are summarised in Table 1, showing the relevant data that combine modifications in human gut microbiota taxa from patients suffering metabolic and endocrine-related diseases and their representative microbial metabolites.

The analysis showed the relevant taxa increased or decreased in specific pathologies together with a metabolite analysis that can serve as a reference. The predominant metabolites variations are related to amino acid metabolism, lipids, and bile acid metabolisms (Figure 3). The main enrichment or depletion of specific taxa is also collected and shown. It is important to highlight the trend of several genera well-known or agreed by the scientific community having a differential role in metabolic diseases when they are depleted or decreased that are: *Akkermansia*, *Faecalibacterium*, *Prevotella*, *Roseburia*; and increased taxa: *Eubacterium*, *Oscillobacter*, *Dorea*, *Ruminococcus*, *Streptococcus*. *Blautia* species are not differentially present in these patients. 

### 2.3. Microbiota Taxa and Metabolite Profiles Linked to Xenobiotic Exposure

Most of the interactions between xenobiotics and gut microbiota and their impact are still unclear. Alterations attributed to the incorporation of xenobiotics remain diffuse as a consequence of the complexity of the interactions between these chemical compounds, gut microbiota resilience and metabolisation capacities, metabolite variations, and regulation of host-microbiota combined metabolisms. Considering the above, information summarised in Figure 4 and Table 2 evaluates the available knowledge that links xenobiotic exposure, metabolite variations, gut microbiota modifications, and metabolic-endocrine diseases. The data used in the analysis show the relevance of the available animal model studies, revealing two main objectives: (i) to visualise which gut taxa are more prone to resist or be decreased by the short-term exposure to specific dietary xenobiotic, and (ii) to observe the regulation or variation of the combined metabolisms of host and microbiota linked to the metabolic health effects.

In this framework, it is also important to compile the available information on interactions between xenobiotics, microbiota taxa and metabolite variations and host status in other well-accepted laboratory in vivo models. Therefore, key complementary information on zebrafish models was compiled (Table 3), in order to establish associations primarily in the context of metabolic, intestinal and hepatic disorders.

### 2.4. NGP Studies for Interventional Metabolic Dysbiosis

*Akkermansia muciniphila*, *Faecalibacterium prausnitzii*, *Bacteroides uniformis*, *Bacteroides acidifaciens*, *Clostridium butyricum*, *Eubacterium hallii*, *Prevotella copri*, and *Christensenella minuta* are members of the gut microbiota that have shown prophylactic and palliative effects in some disorders associated with metabolic and gut dysbiosis [2,3,4,5,6,7,8,9,10]. Taking into account the screening data studies available in Figure 5 and the corresponding extracted information shown in Table 4, strains of NGPs in doses well described seem to be a promising effective therapy in dysbiosis and metabolic alterations. They were able to modulate glucose and lipid homeostasis, and weight balance. The primary metabolites modified by NGPs are short-chain fatty acids and other fatty acids, vitamins, amino acids, polyamines, and bile acid metabolites. Unfortunately, although many of these studies were well designed, they did not include metabolite data analysis.

In mice model studies, *A. muciniphila* has been associated with reduced body weight and reduced weight gain, as well as with reduced fat mass and reduced fat mass gain [38,47,50]. In addition, positive effects on the liver have been associated with this microorganism in mice and rats, including reductions in fatty liver disease, steatosis, and organ injury and an increase in liver function [45,50,51,52]. Moreover, antidiabetic effects of *A. muciniphila* have been reported in multiple studies in murine models, including increased glucose tolerance and blood glucose control and reduced insulin resistance [38,47,50,53].

The analysis of these studies suggests that NGPs may mitigate several metabolic disorders by restoring gut microbiota dysbiosis and modifying specific metabolisms. The altered microbiota taxa cause disturbances mainly on lipid metabolism by increasing substrates for energy metabolism in the liver and peripheral tissues. Data extraction shows that alteration in bile acid metabolism also affects the digestion of dietary lipids in the gut and other signalling functions. The gut microbiota has a deep effect on bile acid metabolism by promoting deconjugation, dehydrogenation, and dehydroxylation of primary bile acids. Other dysbiotic gut bacterial compositional profiles can play an important role in susceptibility to metabolic disorders by affecting amino acid bioavailability to the host.

## 3. Discussion

There is a growing interest and body of knowledge in the analysis of the gut microbiome and its metabolome [56,57], but this is still limited to explaining relevant interconnected physiopathological impacts or finding specific biomarkers. One interesting and integrative programme is the Human Gut Microbiome Atlas (HGMA, available online: https://www.microbiomeatlas.org (accessed on 24 September 2022)) which aims to analyse the human microbiome data from human samples obtained from several diseased and healthy cohorts by integrating metagenomics and other omics data using systems biology. This open-access atlas is updated routinely with the new publicly available gut metagenomics data, including the data from the recently announced one million microbiome project (MMHP, https://db.cngb.org/mmhp/ (accessed on 24 September 2022)) which will provide a comprehensive open-access metagenomics data from multiple research centres. Moreover, similar approaches for distinguishing the microbial metabolites from others (e.g., host, food, or xenobiotics) and exploring their metabolic functions and correlations specifically with the microbiome may improve the efficiency and accuracy of health-disease biomarker discoveries [58]. However, the key point is to integrate more data than those derived from metagenomics analysis.

The focus of this review was the correlation between specific gut microbial taxa and metabolites that are differentially present in metabolic or related-endocrine pathologies. Despite recent evidence that links metabolic disorders with certain gut microbiota populations, currently, it is not possible to classify individuals according to their gut microbial profiles in either eubiotic or dysbiotic states that are associated with homeostasis or disease, respectively. However important data are retrieved for diabetes, NAFLD and cardiovascular patients [59,60,61].

The available information that relates to gut microbiota taxa variations, metabolomic profiles and disease development shows evidence that human gut microbiota may be able to modulate host metabolome and affect final host homeostasis, due to direct and inverse interactions between these three elements. Also, among many other factors, we highlight the important role of xenobiotics as compounds able to affect homeostasis in humans and animals. In this context, it is worth mentioning those cases in which the cumulative exposure to xenobiotics in the host could be related to changes in the microbial populations and in the profile of synthesised metabolites where, ultimately, these compounds could be associated with the long-term deregulation of host metabolic pathways and the development of metabolic-endocrine disorders [11,62,63].

In the case of pesticides, plasticisers such as BPA and its analogues (BPS, BPF, and BPAF), there is evidence of the alterations in microbiota metabolite profiles after exposure. However, the relationship between microbiota populations and the development of metabolic-endocrine disorders is far from being elucidated, due to the lack of currently available information [64,65]. Interestingly, the term reactobiome was proposed for stratifying and unraveling the metabolic features of the gut microbiome that explains resilience and microbiome dysbiosis at a functional level. The authors described five reactotypes with specific amino acid, carbohydrate, and xenobiotic metabolic features [66].

The present study shows that the main bacterial taxa increased in metabolic diseases appear to be species from *Lactobacillus*, *Escherichia*, *Blautia*, *Streptococcus*, and *Klebsiella* genera. In contrast, species decreased are the ones encompassing strains with potential use as NGP: *Faecalibacterium*, *Bifidobacterium*, *Bacteroides*, *Roseburia*, *Akkermansia*, and *Alistipes*. Similarly, results from directed culturing studies adding xenobiotics showed that *Bacillus*, *Clostridium*, *Staphylococcus*, and *Enterococcus* species were isolated when gut microbial samples were exposed to BPA, a xenobiotic with a putative role in dysbiosis and long-term metabolic effects and obesity [67].

*Akkermansia muciniphila*, *Faecalibacterium prausnitzii*, *Bacteroides uniformis*, *Bacteroides acidifaciens*, *Clostridium butyricum*, and *Prevotella copri* are microorganisms associated with the gut microbiota able to exert preventive or palliative effects in a dysbiosis context. The global analysis of the data collected reveals some associations between the administration of these NGPs and an improvement in certain physiological and cognitive parameters in murine models. For instance, Wu et al. [50] demonstrated that the administration of *Akkermansia muciniphila* significantly reduced body weight gain and improved the spatial memory ability of high-fat diet-fed mice. Thus, they could be considered to be NGPs, and their effectiveness should be assessed in future human clinical trials. It is important to retrieve supporting data on the common signature metabolites and microorganisms in well-known microbiota dysbiosis and metabolic diseases (obesity, NAFLD, diabetes, etc.) to search for the NGP that could modulate the pathological phenotypes. The SCFA metabolites produced by NGPs seem to be key players in lipid metabolism as metabolic endpoint modulators [68]. However, it is also important to analyse the regulation of bile acid metabolism by NGPs [69]. The bioavailability of acetyl-CoA derivatives [70] and amino acids to the host by specific microbiota taxa abundance could determine the susceptibility to linked metabolic disorders [71].

In any case, the information available from NGP-human interventions is still very limited. Most studies in this field showed that supplementation with certain NGP could also improve certain metabolic disorders, obtaining beneficial effects in populations with obesity [72]. *A. muciniphila* is one of the better NGP evaluated in the present review and it is considered one of the microorganisms more relevant for its potential clinical applications, due to demonstrated modulatory metabolic capacities, under certain conditions, formula, doses and viability of the bacterial cells. In mice, *A. muciniphila* has been associated with reduction of weight and antidiabetic effects. Importantly, the beneficial anti-diabetic effects of *A. muciniphila* have also been studied in humans more recently, revealing links between the decreased abundance of the organism and impairment of insulin secretion and glucose homeostasis, specifically in lean individuals with type 2 diabetes [73]. Further studies in humans are needed to confirm the beneficial effects that have been reported in animal models and other target species.

In this sense, it is relevant to consider the One Health approach and the complex interconnections existing among human, animal, plant, and environmental health. The reason for this complexity is the multiple exposure pathways, including ingestion, inhalation, dermal contact during food consumption and use of medicines and cosmetics, as well as contact with environmental pollutants that require holistic approaches [74]. Another reason is dealing with the totality of the environmental exposure of the organisms, the exposome, and the related interaction with host genetic factors in common chronic diseases [75]. Recent investigations revealed that microbiota might play an important role in xenobiotic metabolism [76]. Because of these microbiome-xenobiotic interactions, the human gut microbiota may also be a potential source of probiotics for animals and plants, and environmental bioremediation [77]. Eventually, future research will need to overcome the additional hurdle of addressing multiple biological, chemical, and physical hazards affecting multiple target organisms through multidisciplinary and holistic approaches [78].

### 3.1. Limitations of the Study

The impact of dietary xenobiotic exposure on microbiota and metabolic health-disease status has been mainly studied in animal models and in short-term interventions, which makes it difficult to translate and evaluate the impact on humans. Moreover, there are still many studies regarding gut microbiota taxa information only retrieved through metagenomics that lack host-microbiota metabolome profiles, which impair the complete elucidation of the impact of the functionalities in the host. Moreover, enrichment and/or depletion of essential gut microbiota taxa are still controversial in the studies available, which prevent the establishment of well-defined clinical biomarkers and a harmonised impact on health status. In this sense, NGP studies are still too few to demonstrate the validation of its use compared to traditional probiotics. Moreover, NGP face many regulatory issues until they can be used in clinical trials and interventional studies, which, in parallel, slow down the demonstration of their beneficial effects.

### 3.2. Future Perspectives

Probiotics which are traditionally approved and used belong to the lactic acid bacteria group. Current advances in extended omics technologies, not limited to metagenomics, when applied to the study of the human microbiome, have revealed new potential probiotic candidates. Therefore, this field is expanding, and new variants of probiotics are under consideration, such as symbiotics, microbial consortia, or genetically modified microorganisms. For instance, a synbiotic preparation containing *A. muciniphila*, among other probiotics, and inulin as the prebiotic, showed an improvement in glucose levels in type 2 diabetes patients [79]. Regarding microbial consortia, El Hage et al. [80] showed results that support the development of next-generation probiotics composed of multiple bacterial strains when they found that a propiogenic consortia of bacteria was able to restore in vitro propionate concentrations upon antibiotic-induced microbial dysbiosis. Their synthetic community was designed based on core members of the human gut microbiota and with functional redundancy for the production of propionic, thus, taking into account functionality and phylogenetic background. These are promising results to support further clinical trials to test the effectiveness of the synthetic community in treating metabolic syndrome, where the health benefits of the propionate might contribute to ameliorating the disease. These new approaches add complexity to the research question when compared to the traditional single strain probiotics because it is necessary to design/engineer the synthetic community and be able to manage the community efficiently. Therefore, we need to understand the microbial interactions that occur within the synthetic community in order to achieve functional stability and effectiveness toward the restoration of gut diversity and functionality associated with human disorders [81]. It is important to perform multivariate data analysis to obtain predictive effects on the microbiome and metabolism and further related insights from a multi-species comparison. In this sense, several authors suggested performing holistic and clinical studies in zebrafish, where even a moderate coverage of its metabolome may be representative of the global metabolic changes [82].

Future NGPs interventional studies should take advantage of the advances in integrative analysis techniques of multi-omic data, such as machine learning and artificial intelligence, to characterise probiotic effects, including metagenomic, metatranscriptomic, and metabolomic technologies. Integrated approaches may help to identify overexpression or loss of microbiome functions associated with host health or disease and provide further potential for developing NGPs that functionally compensate for the imbalance.

Taking all the above into account and considering the One Health approach mentioned above, it is necessary to apply a more holistic data approach to developing and testing NGPs. This includes the use of ecological and physiological principles in the development of the NGPs, the understanding of the mechanism of action, and integrative analysis of multi-omics data, together with dietary and lifestyle characteristics in large clinical trials.

## 4. Materials and Methods

We performed a comprehensive literature search covering the period from the beginning of January 2017 up to the end of December 2021 using Scopus, Web of Science, and PubMed databases, dividing this review into four main study issues: gut microbial taxa variation and metabolic-endocrine-related diseases; microbiota metabolite modifications and microbiota variation in pathologies; xenobiotics, gut microbiota, and microbial metabolite variations; and Next-Generation Probiotics, using the search strategies explicitly shown in Appendix A and using bibliometrics to carry out a bibliometric analysis of datasets.

The next step involved limiting the search to studies with at least five citations, written in English and excluding duplicates. In this context, 58 clinical studies were selected for data analysis related to “Gut microbial taxa variations and metabolic-endocrine-related diseases”, and 20 studies were selected for data analysis in research on “Microbiota metabolite modifications and microbiota variation in pathologies”. As a result of combining both searches, 7 linked studies were isolated, involving studies on gut microbial taxa variations in metabolic-endocrine-related diseases and metabolomic variations. In addition, 28 studies were included in the research on “Xenobiotics, gut microbiota and microbial metabolite variations”. After excluding articles where one or more of the following fields were unavailable (xenobiotics, gut microbiota variation, metabolite variation, and/or pathology), 5 studies were chosen and summarized after data analysis. Finally, 18 clinical studies were included for further data interpretation in the research on “Next Generation Probiotics”. All steps were presented according to PRISMA Flow Diagrams [83].

## 5. Conclusions

To increase scientific data availability on the interplay between metabolic and molecular pathways involving xenobiotic exposure and their biodegradation, gut microbiota taxa and metabolite modification needs to be studied continuously, using improved methods. It will allow for the development of new biological-based treatments for mitigating metabolic disorders and diseases.Relevant modifications of potential signature metabolites mediated by targeted microbiota taxa belong to lipid, bile acid, acetyl-CoA, and amino acid metabolisms.The selection and application of appropriate NGPs from healthy microbiota, after elucidating their abundance, functionality, and key molecular mechanisms, seems to be a promising strategy to potentially restore the homeostasis of the intestinal microbiota, taking into account food safety and risk assessment studies and their clinical impact in murine models and subsequently validation in human studies.Exploring the uses of NGPs in animals, plants, and/or bioremediation following the preliminary steps of the *One Health* approach before clinical administration can overcome many safety issues posed by the use of new beneficial microbes in humans. Moreover, it could demonstrate the metabolic potential of NGPs to help refine doses and formulations.

## Figures and Tables

**Figure 1 ijms-23-12917-f001:**
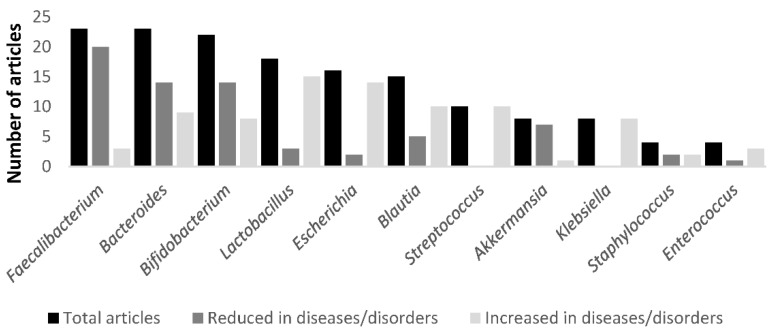
Analysis of main taxa variability in relation to metabolic disorders. Differentially reduced genera: *Faecalibacterium*, *Bacteroides*, *Roseburia*, *Akkermansia*, and *Alistipes* and differentially increased genera: *Lactobacillus*, *Escherichia*, *Blautia*, *Streptococcus*, and *Klebsiella*.

**Figure 2 ijms-23-12917-f002:**
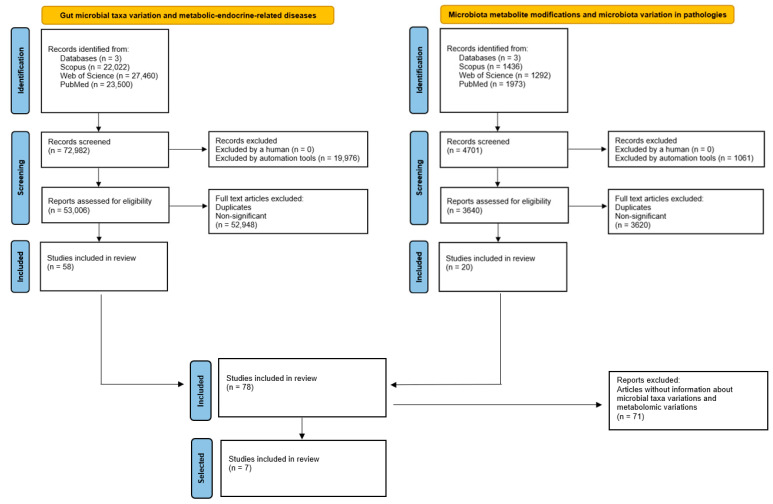
Combined PRISMA diagrams for gut microbial taxa variation in metabolic diseases and the associated microbiota metabolite profiles.

**Figure 3 ijms-23-12917-f003:**
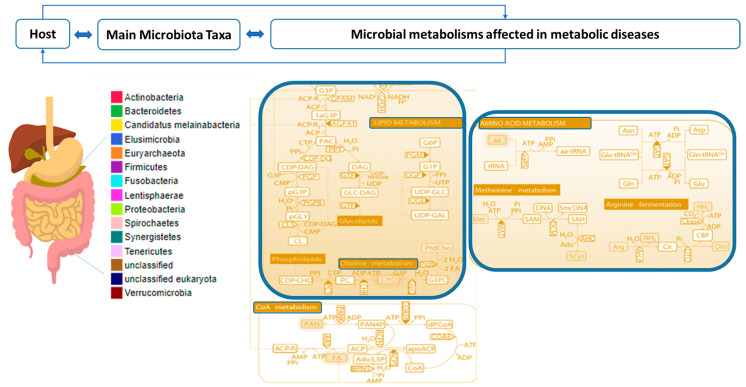
Scheme of gut microbiota taxa and metabolites analysis. Main microbial metabolisms affecting metabolic diseases (lipid, bile acids, acetyl-CoA and amino acids metabolisms).

**Figure 4 ijms-23-12917-f004:**
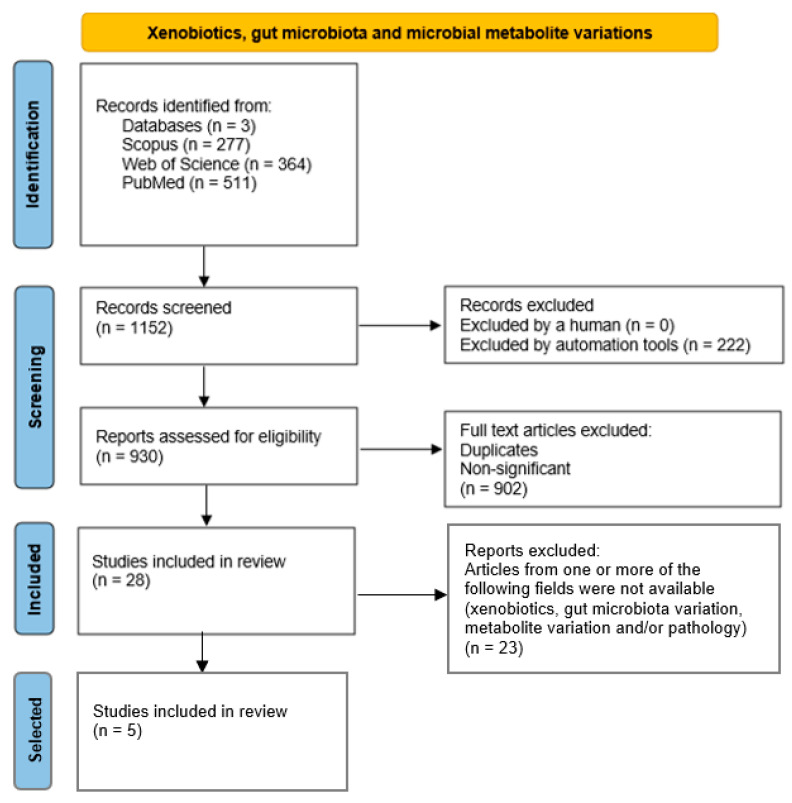
PRISMA diagram about xenobiotics, gut microbiota variations, metabolite modification and host status.

**Figure 5 ijms-23-12917-f005:**
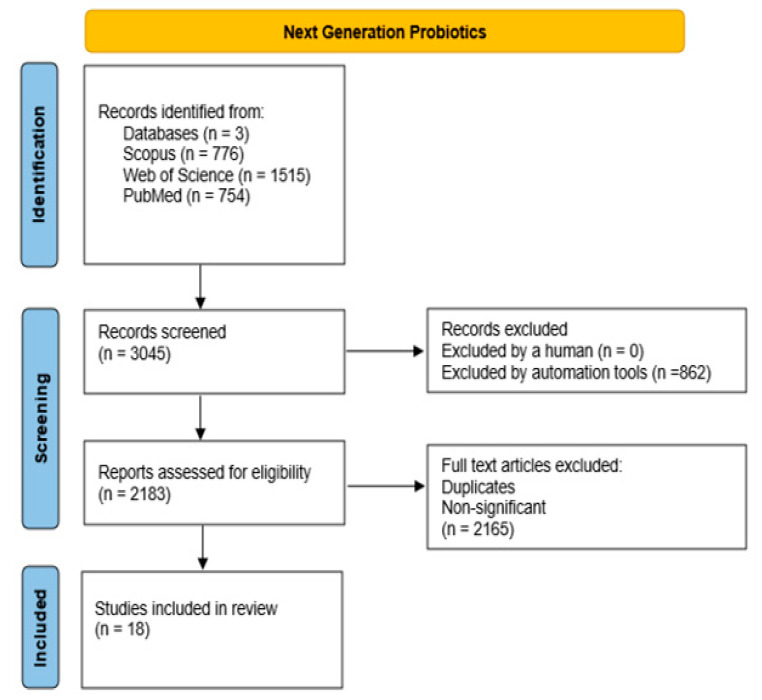
PRISMA diagram about NGP information available and data analysis of selected studies.

**Table 1 ijms-23-12917-t001:** Microbiota taxa and metabolite variations found in patients suffering metabolic diseases.

Ref.	Clinical Traits	Microbiota Taxa Modification	Metabolite Modifications—Pathways
[13]	*n* = 115; HC *n* = 54; OB *n* = 8; NAFLD *n* = 27; NASH *n* = 26	↑ *Bradyrhizobium*, *Anaerococcus*, *Peptoniphilus*, *Propionibacterium acnes*, *Dorea*, and *Ruminococcus* ↓ *Oscillospira* in NAFLD, NASH and OB vs. HC	2-Butanone, 4-Methyl-2-pentanone **Ketone pathways**
[14]	*n* = 1280; LN-NonT2D *n* = 633; OB-NonT2D *n* = 494; OBT2D *n* = 153	↓ *Akkermansia*, *Faecalibacterium prausnitzii*, *Oscillibacter*, and *Alistipes* in OB	Indolepropionate, 2-Methylbutyrylcarnitine, Valine, Isovalerate, Glutamine, Tyrosine, 3-Phenylpropionate, Phenylalanine, Oxalate, *N*1-Methyl-2-pyridone-5-carboxamide, Docosapentaenoate, 1-Stearoyl-GPE (18:0), 10-Heptadecenoate, 1-Arachidonoyl-GPI (20:4), Inosine, Glycylvaline, Citrulline, Gamma-CEHC, 1-Linoleoyl-GPC (18:2), Adrenate (22:4n6), Epiandrosterone sulphate, 2-Linoleoyl-GPC (18:2), 1-Oleyl-GPC (18:1), 1-Dihomo-linoleoyl-GPC (20:2), Cinnamoylglycine **Amino acid and phospholipid metabolism pathways**
[15]	*n* = 100; HC *n* = 35; T2D+ *n* = 49; T2D- *n* = 16	↑ *Coprococcus 1* ↓ *Bacteroides* and *Prevotella* in T2D+ and T2D- vs. HC; ↑ *Parasutterella* in T2D+ vs. HC; ↑ *Blautia* and *Eubacterium hallii* group in T2D- vs. HC	HDL cholesterol, LDL cholesterol, Acetate, Butyrate, Linoleic acid, Palmitoylcarnitine, Lysophosphatidylcholine (18:2), Phosphatidylcholine (16:0/17:0), Diacylglycerol (15:0/18:3), Diacylglycerol (15:0/20:3), Glycoursodeoxycholic acid, Chenodeoxyglycocholate, Glycocholic acid, Cholic acid **Lipid metabolism, bile acid metabolism and cholesterol pathways**
[16]	*n* = 69; HC *n* = 40; Non-PN SBS *n* = 5; SBS I *n* = 10; SBS II *n* = 14	↑ *Lactobacillus* and *Klebsiella* ↓ *Coprococcus*, *Faecalibacterium*, *Lachnospira*, and *Ruminococcus* in SBS patients; ↓ *Blautia*, *Bacteroides*, *Odoribacter*, *Oscillospira*, *Prevotella*, *Roseburia*, and *Sutterella* in SBS I and SBS II; ↑ *Streptococcus* and *Staphylococcus* in SBS I	Butanoic acid, Pentanoic acid, 1-Nonanol, p-Cresol, Geranil acetone, γ-Undecalactone, Indole, Phenol, Decanoic acid, Dodecanoic acid, Nonanal, Octanal, Hexanal, 2-pentyl furan, Lythocholic acid, Taurocholic acid, Chenodeoxycholic acid Deoxycholic acid, Glycodeoxycholic acid, Cholic acid, Glycocholic acid, Glycochenodeoxycholic acid **Volatile organic compounds and bile acid metabolism pathways**
[17]	*n* = 155; Non-IBD *n* = 34; CD *n* = 68; UC *n* = 53	↑ *Eubacterium ventriosum*, *Coprococcus catus*, *Roseburia hominis*, *Dorea longicatena*, *Eubacterium hallii*, *Eubacterium siraeum*, *Alistipes shaii*, *Alistipes putredinis*, *Alistipes finegoldii*, *Roseburia inulinivorans*, *Roseburia intestinalis*, *Faecalibacterium prausnitzii*, *Eubacterium eligens*, *Bacteroidales bacterium ph8*, *Alistipes indistinctus*, *Alistipes senegalensis*, *Ruminococcus callidus*, *Holdemania filiformis*, *Fordonibacter pamelaeae*, *Lachnospiraceae bacterium 1*, *Adlercreutzia equolifaciens*, and *Alistipes onderdonkii* in Non-IBD controls; ↑ Unclassified *Roseburia* species in CD and UC; *↑ Bifidobacterium breve* and *Clostridium symbiosum* in UC; ↑ *Blautia producta*, *Lactobacillus gasseri*, *Enterococcus faecium*, *Lachnospiraceae bacterium 2*, *Clostridium clostridioforme*, *Ruminococcus gnavus*, and *Escherichia coli* in CD	Caprylic acid, Carnosol, Urobilin, Pipecolic acid, 4-Methylcatechol, 2-Hydroxyhexadecanoate, Cholestenone, 5ɑ-Cholestanol, Dodecanedioic acid, Caproic acid, Hydrocinnamic acid, 3-Methyladipate-pimelate, Undecanedionate, Azelaic acid, 2-Hydroxyphenethylamine, Linoleoyl ethanolamide, Palmitoylethanolamide, Docosapentaenoic acid, Eicosatrienoic acid, Taurine, *N*-Acetylputrescine, ADMA, Cholate, Chenodeoxycholate, Phytosphingosine, C 18:0 CE, C14 carnitine, C3-DC-CH3 carnitine **Bile acid metabolism pathways**
[18]	*n* = 196; HC *n* = 41; pHT *n* = 56; HT *n* = 99	↑ *Prevotella* and *Klebsiella* in pHT or HT; ↑ *Porphyromonas* and *Actinomyces* in HT; ↓ *Faecalibacterium*, *Oscillibacter*, *Roseburia*, *Subdoligranulum*, *Blautia*, *Bifidobacterium*, *Coprococcus*, *Butyrivibrio*, *Eggerthella*, *Streptococcus*, and *Akkermansia* in pHT and HT	Hippurin-1, Trichloroethanol glucuronide, PS(O-18:0/0:0), LysoPC(18:2), S-Carboxymethyl-L-cysteine, Pyridine, LysoPC (22:5), 3-Keto stearic acid, Petunidin 3-rhamnoside 5-glucoside, Nɑ-Acetyl-L-arginine, 9,10-Dichloro-octadecanoic acid, PA(12:0/0:0) **Glucuronide detoxification and antioxidant pathways**
[19]	*n* = 201; HC *n* = 40; CAD *n* = 161	↑ *Actinomyces*, *Haemophilus*, *Granulicatella*, *Weissella*, *Veillonella*, *Streptococcus*, *Klebsiella*, *Rothia*, *Enterococcus* (CAG17); ↓ *Faecalibacterium*, Lachnospiraceae, *Roseburia*, *Oscilibacter* (CAG4); Lachnospiracea incertae sedis, *Ruminococcus 2*, *Dorea*, *Blautia*, *Clostridium XVIII* (CAG14); *Anaerostipes*, *Blautia*, *Lactobacillus*, *Fusicatenibacter*, *Clostridium XIVa*, *Gemella*, *Bifidobacterium*, *Saccharibacteria genera incertae sedis* (CAG15); *Roseburia*, Lachnospiracea incertae sedis, *Clostridium XIVb*, *Parasutterella*, *Butyricicoccus* in CAD	Steroids, Sphingolipids, Phosphatidylethanolamine, Phosphatidylcholine, Ceramides, Glycerophospholipid, Fatty acyls, Carboxylic acids, Benzene/derivatives, Fatty acyl carnitines, Prenol lipids, Glycerolipids, Potassium chloride, Addictives/ingredients, Taurine, Aminoacids (L-Leucine) **Amino acid and lipid metabolism pathways**

↑ Taxa increased; ↓ Taxa decreased; CAD: coronary artery disease; CAG: co-abundance group; CD: Crohn’s disease; HC: healthy control; HT: hypertension; IBD: inflammatory bowel disease; LN: lean; NAFLD: non-alcoholic fatty liver disease; NASH: non-alcoholic steatohepatitis; Non-PN SBS: parenteral nutrition-independent short bowel syndrome; OB: obese; pHT: prehypertension; SBS I: parenteral nutrition-dependent short bowel syndrome I; SBS II: parenteral nutrition-dependent short bowel syndrome II; T2D: type 2 diabetes; T2D+: type 2 diabetes with chronic complications; T2D-: type 2 diabetes without chronic complications; UC: ulcerative colitis.

**Table 2 ijms-23-12917-t002:** Relationship between xenobiotics exposure, microbiota taxa and metabolite variations and host status in mice model.

Ref., Xenobiotic, Biosample	Microbiota Taxa Modification	Metabolite Modification	Health Effects
[20] **Chlorfenapyr; acetamiprid; and chlorfenapyr + acetamiprid** Kunming mice *n* = 60; CK *n* = 20; C *n* = 10; A *n* = 10; AC *n* = 10; N = 10 Faeces and serum	*↑ Helicobacter*, *Desulfovibrio*, *Oscillibacter*, *Intestinimonas*, *Roseburia*, *Lachnoclostridium*, *Ruminiclostridium*, and *Butyricimonas* in chlorfenapyr *↓ Lactobacillus*, *Bacteroides*, *Parasutterella*, *Erysipelatoclostridium*, *Enterorhabdus*, *Alloprevotella*, and *Enterococcus* in chlorfenapyr ↑ *Lactobacillus* and *Marvinbryantia* in acetamiprid ↓ *Muribaculum*, *Parabacteroides*, and Unidentified Clostridiales in acetamiprid	↑ Trimethylamine-*N*-oxide, cholic acid derivative, 5-β-cholanoic acid, 3-β-hydroxy-5-cholenoic acid, 7-ketodeoxycholic acid, avicholate, methylcholate, and uric acid in C, A, and AC (Faeces) ↓ Free fatty acid in C, A and AC (Serum) ↑ Betaine in A and AC (Faeces) *↑* Long-chain free fatty acids and esters in A and C (Faeces) ↓ Phosphatidylcholine and phosphatidylethanolamine in A and C (Serum) *↑* 5-Hydroxyinoleacetic acid and indole-2-carboxylic acid in A (Faeces) ↑ Free fatty acid, *N*-acetyl-tryptophan, and *N*-acetyl-phenylalanine in A (Serum) *↓* 3-(Aminomethyl)-indole, indoline, and indolemethanamine in C (Faeces) ↑ Tryptophan in C (Serum)	Glucose homeostasis
[21] **2,2′,4,4′-Tetrabromodiphenyl ether** ICR mice *n* = 36; ND+V *n* = 6; ND+L-BDE *n* = 6; ND+H-BDE *n* = 6; HFD + V *n* = 6; HFD+L-BDE *n* = 6; HFD+H-BDE *n* = 6 Faeces and serum	↑ *Parasutterella* and *Gemella* in ND+L-BDE ↓ *Christensenellaceae* R-7 group, *Atopostipes*, Family *XIII UCG-001*, and *Bacillus* in ND+L-BDE ↑ *Candidatus Saccharimonas*, *Ruminococcaceae UCG-013*, *Staphylococcus*, *Eubacterium nodatum* group, *Gemella*, *Corynebacterium 1*, and *Paenalcaligenes* in ND+H-BDE↑ *Staphylococcus* in HFD+L-BDE ↓ *Bacteroides*, *Ruminiclostridium 9*, *Helicobacter*, *Alloprevotella*, *Oscillibacter*, *Christensenellaceae R-7* group, *Ruminiclostridium 5*, *Odoribacter*, *Ruminococcaceae UCG-010*, and *Rikenella* in HFD+L-BDE ↓ *Turicibacter*, and *Anaerotruncus* in HFD+L-BDE and HFD+H-BDE ↑ *Dorea*, *Lactococcus*, and *Eubacterium nodatum* group in HFD+H-BDE ↓ *Ruminococcaceae UCG-014*, *Ruminococcaceae UCG-009*, *Candidatus Saccharimonas*, *Ruminiclostridium 5*, and *Family XIII UCG-001* in HFD+H-BDE	↑ Bile acids, succinate, taurine, glycine, α-glucosa, β-glucose, arabinose, and galactose in ND-BDE (Faeces) ↓ Methionine in ND-BDE (Faeces) ↑ Bile acids, choline, α-ketoglutarate, and α-glucose in HFD-BDE (Faeces) ↓ Propionate and β-glucose in HFD-BDE (Faeces) ↑ Pyruvate, lactate, phosphoric acid, glutamine, ornithine, 3-hydoxybutyric acid, isoleucine, and octadecanoic acid in HFD-BDE (Serum) ↓ Palmitelaidic acid and uric acid in HFD-BDE (Serum)	Obesity Steatosis Glucose homeostasis
[22] **Tebuconazole** ICR mice *n* = 24; Control *n* = 8; L-TEB *n* = 8; H-TEB *n* = 8 C57BL/6 mice *n* = 16; Control *n* = 8; TEB+DSS *n* = 8 Serum	*↑ S24-7*, *Coprococcus*, and *Akkermansia* in H-TEB *↓ Clostridiales*, *Ruminococcaceae*, *Ruminococcus*, *Oscillospira*, *Mucispirillum*, *Rikenellaceae*, and *Dehalobacterium* in H-TEB *↑ Rikenellaceae*, *Akkermansia*, and *Bilophila* in TEB+DSS *↓* *S24-7* in TEB+DSS	*↑* α-Glucose, β-glucose, taurine, leucine, lysine, alanine, creatine, glutamine, and glutamate in H-TEB *↓* Lipid, lactate, acetate, and choline in H-TEB *↑* α-Glucose, β-glucose, taurine, leucine, lysine, alanine, and creatine in TEB+DSS *↓* Lipid, lactate, acetate, and choline in TEB+DSS	Colitis
[23] **Di(2-ethylhexyl) phthalate (****DEHP)** C57BL/6J mice *n* = 24; Control *n* = 8; L-DEHP *n* = 8; H-DEHP *n* = 8 Liver	↑ *Streptococcus* and *Butyrivibrio* ↓ *Lactobacillus*	↑ Stearic acid (18:0), linoleic acid (18:2n6), α-linolenic acid (18:3n3), γ-linolenic acid (18:3n6), arachidonic acid (20:4n6), eicosapentaenoic acid (20:5n3), docosaexaenoic acid (22:6n3), glycerophosphoserine, and glycerophosphoglycerol in DEHP ↓ Glycerophosphocholine, glycerophosphoinositol, lysophosphosphatidylethanolamine, lysophosphatidylcholine, phosphatidylethanolamine, and sphingomyelin in DEHP	Obesity
[24] **Carbendazim** C57BL/6 mice *n* = 32; Control *n* = 8; L-CBZ *n* = 8; M-CBZ *n* = 8; H-CBZ *n* = 8 Faeces	↑ Actinobacteria ↓ Bacteroidetes and Verrucomicrobia	*↑* Propionate and butyrate in CBZ *↓* Acetate in CBZ	Hyperlipidaemia

↑ Taxa increased; ↓ Taxa decreased; A: acetamiprid; BDE: 2,2′,4,4′-tetrabromodiphenyl ether; C: chlorfenapyr; CBZ: carbendazim; CK: control check; DEHP: di(2-ethylhexyl) phthalate; DSS: dextran sulphate sodium; H-BDE: high-dose 2,2′,4,4′-tetrabromodiphenyl ether; H-CBZ: high-dose carbendazim; H-DEHP: high-dose di(2-ethylhexyl) phthalate; HFD: high fat diet; H-TEB: high-dose tebuconazole; L-BDE: low-dose 2,2′,4,4′-tetrabromodiphenyl ether; L-CBZ: low-dose carbendazim; L-DEHP: low-dose di(2-ethylhexyl) phthalate; M-CBZ: median-dose carbendazim; N: only water; ND: normal diet; TEB: tebuconazole; V: vehicle.

**Table 3 ijms-23-12917-t003:** Relationship between xenobiotics exposure, microbiota taxa and metabolite variations and host status in zebrafish model.

Ref., Xenobiotic, Doses	Metabolite Modifications	Gut Microbiota Taxa Modification	Health Status
[25] **Bisphenol A** BPA (2 and 20 µg/L)	↑ Serotonin in BPA-female ↓ Serotonin in BPA-male	↑ *Hyphomicrobium* in BPA	Intestinal health and oxidative stress
[26] **Bisphenol F** BPF (0.5, 5, and 50 µg/L)	↑ Glutamate, arginine, succinate, D-serine, L-tyrosine, adenine, inosine, hypoxanthine, xanthine, and guanine in BPF	↑ *Ralstonia* in BPF ↓ *Gemmobacter* in BPF	Hepatic fibrosis and steatosis
[27] **Bisphenol F** BPF (2, 20, and 200 μg/L)	L-glutamine, L-tyrosine, L-tryptophan, L-glutamate, L-leucine, L-isoleucine, and L-proline in BPF	↑ *Microbacterium*, *Mycobacterium*, *Pseudomonas*, and uncultured bacteria in BPF ↓ *Burkholderia–Caballeronia–Paraburkholderia*, *Bifidobacterium*, *Cetobacterium*, and *Halomonas* in BPF	Neurotoxicity
[28] **Chlorpyrifos** CPF (30, 100, and 300 µg/L)	Celobiose, α-tocopherol, gentiobiose, β-mannosylglycerate, glucose-6-phosphate, gluconic acid, isomaltose, 3-hydroxyflavone, L-malic acid, glucose, mannose, 3-hydroxypropionic acid, maltose, lactic acid, 4-aminobutyric acid, phenyl β-D glucopyranoside, *N*-acetyl- β-D-mannosamine, fructose, heptadecanoic acid, neohesperidin, 2-monopalmitin, adrenosterone, 7-α-hydroxycholesterol, ethanolamine, glycerol, D-glyceric acid, 2-hydroxyvaleric acid, 4-cholesten-3 one 4, ergosterol, myristic acid, L-4-hydroxyphenylglycine, 3-hydroxy-L-proline, O-methylthreonine, cycloleucine, picolinic acid, shikimic acid, glutamic acid, β-alanine, oxoproline, serine, urail, phenanthrene, abietic acid, pantothenic acid, and cis-gondoic acid in CPF	↑ β-Proteobacteria in CPF ↓ α-Proteobacteria and γ-Proteobacteria in CPF	Hepatic metabolism and oxidative stress
[29] **Chlorpyrifos** **Micro-Siced Polystyrene** CPF (0.02, 0.2, 2, 20, and 200 μg/g) mPS (50 and 500 μg/g)	Chlorpyrifos-oxon and mPS-adsorbed chlorpyrifos (MIX1 and MIX2)	↑ *Xanthobacter* and *Methylobacterium-Methylorubrum* in CPF ↓ *ZOR0006*, *Chitinibacter*, *Paucibacter*, *Rhodococcus*, and *Cetobacterium* in CPF ↑ *Vibrio*, *Rhodococcus*, and *unclassified_f_Rhizobiaceae* in chlorpyrifos-loaded mPS ↓ *Aeromonas*, *Cetobacterium*, *Chitinibacter*, and *Flavobacterium* in chlorpyrifos-loaded mPS	Hepatic metabolism, intestinal health, oxidative stress and locomotivity
[30] **Propamocarb** PM (100 and 1000 μg/L)	↑ Sucrose-6-phosphate, 1-kestose, glucose-6-phosphate, glycerol, lactic acid, thymine, ribitol, ribulose-5-phosphate, oxoproline, orotic acid, pyridoxine, glutamic acid, and succinic acid in PM ↓ 6-methylmercaptopurine, 3-aminoisobutyric acid, glutamine, lysine, isoleucine, L-allothreonine, glycine, serine, isocitric acid, fumaric acid, L-malic acid, aspartic acid, phenylalanine, valine, threonine, and methionine in PM	*Deefgea*, *Flavobacterium*, *Cupriavidus*, *Megamonas*, *Sediminibacterium*, *Acinetobacter*, *Cetobacterium*, and *Shewanella* in PM	Hepatic metabolism
[31] **Carbendazim** CBZ (30 and 100 μg/L)	↓ Glucose and pyruvate in CBZ	*↑ Phascolarctobacterium*, *Macellibacteroides*, *Shewanella*, *Faecalibaculum*, *Turicibacter*, *[Eubacterium]_xylanophilum_group*, and *Crenobacter* in CBZ *↓ Erysipelatoclostridium*, *Chryseobacterium*, *Bryobacter*, *Gemmobacter*, *Caulobacter*, *Nicotiana_otophora*, *Pelomonas*, and *Alistipes* in CBZ	Hepatic metabolism
[32] **Difenoconazole** DFZ (0.4, 1, and 2 mg/L)	↑ Triglycerides and malondialdehyde in DFZ	*↑ Plesiomonas*, *Aeromonas*, *Firmicutes*, *Ochrobactrum*, *Rhodobacteraceae*, *Enterobacteriaceae*, *Comamonadaceae*, *Gemmobacter*, *Shewanella*, and *Bacteroides* in DFZ ↓ *Cetobacterium* in DFZ	Hepatic metabolism and intestinal health
[33] **Imazalil** IMZ (100 and 1000 μg/L)	↑ Cellobiose, maltose, maltotriose, L-threose, sucrose-6-phosphate, trehalose-6-phosphate, 3-aminoisobutyric acid, ribose-5-phosphate, 6-phosphogluconic acid, pyrubic acid, citramalic acid, cholesterol, palmitic acid, phytanic acid, heptadecanoic acid, stearic acid, arachidonic acid, and myristic acid in IMZ ↓ AMP, dTMP, glutamine, alanine, serine, threonine, isoleucine, proline, valine, malate, pantothenic acid, taurine, orotic acid, and lauric acid in IMZ	↑ Fusobacteria and Firmicutes in IMZ ↓ Bacteroidetes and Proteobacteria in IMZ	Hepatic metabolism and intestinal health
[34] **Di-2-(ethylhexyl) phthalate** DEHP (10, 33, and 100 μg/L)	↑ Triglycerides, pyruvate, and glucose in DEHP-female ↑ Triglycerides, pyruvate, and non-esterified fatty acids in DEHP-male	↑ Proteobacteria and Firmicutes in DEHP-female ↓ Fusobacteria, Bacteroidetes, and Actinobacteria in DEHP-female ↑ Proteobacteria and Bacteroidetes in DEHP-male ↓ Fusobacteria, Firmicutes, and Actinobacteria in DEHP-male	Intestinal health and obesity
[35] **Di-2-(ethylhexyl) phthalate** DEHP (3 mg/kg)	↑ Thioguanine in DEHP-female ↓ D-fructose-6-phosphate in DEHP-female ↑ Choline, ethanolamine, and thioredoxin in DEHP-male ↓ L-Glutamine, L-citruline, and folic acid in DEHP-male	↑ Fusobacteria, Bacteroidetes, and Verrucomicrobia in DEHP	Intestinal health
[36] **Polybrominated Diphenyl Ethers** PBDE mixture (DE-71) (5 ng/L)	↓ Serotonin in DE-71	↑ *Streptococcus*, *Bacillus*, *Helicobacter*, *Moraxella*, *Fischerella*, *Xanthomarina*, and *Tannerella* in DE-71 male ↓ *Lactobacillus*, *Chlamydia*, *Glutamicibacter*, *Paenibacillus*, *Olsenella*, *Ralstonia*, *Mycoplasma*, *Mucilaginibacter*, *Ruminiclostridium*, *unclassified Firmicutes sensu stricto*, *Eubacterium*, *Prevotella*, and *Fusobacterium* in DE-71 male ↑ *Streptococcus*, *Lactobacillus*, *Haemophilus*, *Leptospira*, *Paenibacillus*, *Staphylococcus*, *Helicobacter*, *Mucilaginibacter*, *Neisseria*, *Pseudomonas*, *Aeromonas*, and *Listeria* in DE-71 female ↓ *Acinetobacter*, *Bacillus*, *Glutamicibacter*, *Mycoplasma*, *Ruminiclostridium*, *unclassified Lachnospiraceae*, *unclassified Firmicutes sensu stricto*, *Eubacterium*, *Moraxella*, *Fischerella*, *Fusobacterium*, *Plesiomonas*, *Burkholderia*, *Xanthomarina*, *Xenorhabdus*, *Nonomuraea*, *Alicyclobacillus*, and *Mannheimia* in DE-71 female	Intestinal health and oxidative stress
[37] **Methylparaben** MeP (1, 3, 10 μg/L)	↑ Serotonin in MeP-male ↓ Serotonin in MeP-female	↑ *Mycoplasma* and *Cetobacterium* in MeP	Intestinal health and oxidative stress

↑ Metabolite or Taxa increased; ↓ Metabolite or Taxa decreased.

**Table 4 ijms-23-12917-t004:** NGP strains suitable for therapeutic use in dysbiosis and metabolic disorders with or without metabolite data analysis and health impact.

Ref., NGP Strain, Doses, Target	Metabolite Modifications	Health Effects
[38] *Akkermansia muciniphila* (ATCC BAA-835), 2 × 10^8^ CFU/200 µL, C57BL/6 mice	↑ α-Tocopherol and β-sitosterol ↓ Citrulline and ornithine **Vitamin and Amino acid metabolites**	↑ Glucose tolerance ↓ Weight gain ↓ Fat mass
[39] *Akkermansia* *muciniphila,* 1 × 10^8^ to 10^9^ CFU/100 µL, C57BL/6 mice	↑ *N*1, *N*12-Diacetylspermine, *N*1-acetylspermine, *N*1-acetylspermidine, *N*1, *N*8-diacetylspermidine, spermidine, ornithine, putrescine, acetate, propionate, butyrate, 2-hydroxybutyrate, ketoisovaleric acid, ketoisocaproic acid, ferulic acid, 2-hydroxy-3-methylbutyric acid, deoxycholic acid, hyodeoxycholic acid, murideoxycholic acid, hyocholic acid, lithocholic acid, Ω-muricholic acid, taurodeoxycholic acid, tauro-muricholic acid, taurohyodeoxycholic acid, tauroursodeoxycholic acid, chenodeoxycholic acid, β-muricholic acid, and ursodeoxycholic acid **Polyamine metabolites, short-chain fatty acids and bile acid metabolites**	↑ Pleiotropic metabolic effects supporting gut homeostasis and host health. ↑ Antiaging and anticancer effects
[40] *Faecalibacterium prausnitzii* (ATCC 27766), 2 × 10^8^ CFU/220 µL, C57BL/6N mice	↑ Dihomo-γ-linolenic acid (20:3n6) ↓ Stearic acid (18:0), arachidonic acid (20:4n6), eicosapentaenoic acid (20:5n3), and docosahexanoic acid (22:6n3) ↓ Palmitic acid (16:00) ↓ Linoleic acid (18:2n-6), α-linoleic acid (18:3n3), and eicosapentaenoic acid (20:5n3) **Fatty acid and lipid metabolites**	↑ Weight gain ↓ Hepatic injury
[41] *Bacteroides uniformis* (CECT 7771), 5 × 10^7^ CFU/day, C57BL/6J mice	↑ Butyrate, stearic acid (18:0), and arachidic acid (20:0)↓ Monounsaturated fatty acids, diunsaturated fatty acids, and polyunsaturated fatty acids **Short-chain fatty acids** **and fatty acid lipid metabolites**	↑ Glucose tolerance ↓ Weight gain ↓ Serum cholesterol
[42] *Bacteroides acidifaciens* (JCM10556), 5 × 10^9^ CFU/100 µL, C57BL/6 mice	↑ Cholate and taurine ↓ Butyrate **Short-chain fatty acids** **and bile acid metabolites**	↓ Weight gain ↓ Fat mass ↓ Insuline resistance
[43] *Clostridium butyricum* (CGMCC0313.1), 2.5 × 10^8^ CFU/kg/day, NOD mice	↑ Butyric acid **Short-chain fatty acids** **metabolites**	↓ Diabetes ↓ Diabetes-induced energy metabolic dysfunction
[44] *Prevotella copri* (DSM 18205), 5 × 10^8^ CFU, GK/Ox rats	↑ Cholic acid, allolithocholic acid, chenodeoxycholic acid, and ω-muricholic acid **Total and primary bile acids metabolites**	↑ Glucose tolerance
**Ref., NGP Strain, Doses, Target**	**Metabolite Modifications**	**Health Effects**
[45] *Akkermansia muciniphila* (ATCC BAA-835), 1 × 10^8^–10^9^ CFU/mL, C57BL/6N mice	Not determined	↓ Fatty liver disease
[46] *Akkermansia muciniphila* Muc^T^ (ATTC BAA-835), 2 × 10^8^ CFU/200 µL, Ercc1^−/^^Δ^^7^ mice	Not determined	↑ Restoration of mucus layer
[47] *Akkermansia muciniphila* Muc^T^ (ATTC BAA-835), 2 × 10^8^ CFU/150 µL, C57BL/6J mice	Not determined	↑ Glucose tolerance ↓ Body weight ↓ Fat mass gain ↓ Insuline resistance
[48] *Akkermansia muciniphila* Muc^T^ (ATTC BAA-835), 1 × 10^8^ CFU/200 µL, C57BL/6 mice	Not determined	↓ [Cd] in kidney
[49] *Akkermansia muciniphila* strain *(*139) and (ATCC^T^), 2 × 10^8^ CFU/200 µL, C57BL/6 mice	Not determined	↓ Colitis
[50] *Akkermansia muciniphila*^sub^, 1 × 10^9^ CFU/200 µL, C57BL/6 mice	Not determined	↑ Blood glucose control ↓ Weight gain ↓ Liver steatosis ↓ Memory decay
[51] *Akkermansia muciniphila* Muc^T^ (CCUG 64013), 1.5 × 10^9^ CFU/200 µL, C57BL/6 mice	Not determined	↑ Restoration of mucus layer ↓ Hepatic injury, steatosis
[52] *Akkermansia muciniphila* (DSM 22959), 5 × 10^6^–5×10^8^/500 µL, SD rats	Not determined	↑ Liver function
[53] *Akkermansia muciniphila* (GP01), 5 × 10^9^ CFU/200 µL, APP/PS1 mice	Not determined	↑ Glucose tolerance ↓ Hyperlipidemia ↓ Hepatic steatosis ↓ Intestinal barrier dysfunction
[54] *Bacteroides uniformis* (CECT 7771), 1 × 10^8^ CFU, C57BL/6 mice	Not determined	↓ Weight gain ↓ Cholesterol, triglycerides, glucose
[55] *Bacteroides uniformis* (CECT 7771), 1 × 10^8^–1 × 10^10^ CFU/day, Wistar rats	Not determined	↓ Hepatic alanine aminotransferase

↑ Metabolite or Taxa increased; ↓ Metabolite or Taxa decreased;

## Data Availability

Not applicable.

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
