# Peer review of "Exploring Next Generation Probiotics for Metabolic and Microbiota Dysbiosis Linked to Xenobiotic Exposure: Holistic Approach"

_ijms, 2022, doi:10.3390/ijms232112917_

Round 1

Reviewer 1 Report

The Review written by Torres-Sanchez et al is a comprehensive overview of the topic and should be accepted for publication.

Author Response

REVIEWER 1

The Review written by Torres-Sanchez et al is a comprehensive overview of the topic and should be accepted for publication.

AUTHORS REPLY

THANKS SO MUCH FOR THE POSITIVE EVALUATION OF THE MANUSCRIPT

Reviewer 2 Report

1) The review is interesting and relevant. I suggest making clear the importance of the work / the subject addressed.

2) I don't like the title and I believe it may confuse the reader. Define "endocrine disorders". It's very broad. In addition, the authors mention the taxonomy ("taxa") of the Microbiome, while the most coherent term would be "Microbiota". It would be good to review the term "Microbiome taxa" throughout the text and make changes to the title.

3) The discussion is portrayed in a superficial way, often even non-specific, with little grounded theoretical reference (references); it is possible to improve. The "Next Generation Probiotics (NGP)" mentioned are little explored and this is an extremely important content that maybe could have an improved discussion. The discussion is relatively short, it can be further explored.

4) The authors reviewed the changes in the diversity of the intestinal microbiota from a taxonomic point of view, as well as the profile of metabolites produced, both under the influence of xenobiotics or under conditions of metabolic disorders. In addition, they bring the proposal of the use of probiotics as a therapeutic alternative. However, I suggest including the subject "Future Perspectives" at the end of the manuscript, seeking to further explore what, which and how NGPs can be a promising alternative for many diseases (a fact that is unclear), also exploring the clinical application of this study.

5) Do the authors consider that there are any limitations of the study? If yes, include it in the manuscript.

6) The conclusion is generic, tends towards a more superficial content, that is, non-specific.

Author Response

REVIEWER 2

AUTHORS REPLY

  • The review is interesting and relevant.

THANKS SO MUCH FOR THE POSITIVE CRITICISMS

  • I suggest making clear the importance of the work / the subject addressed.

The abstract and Introduction have been modified according to this suggestion.

Abstract: Main paragraph/s adressing the issue.

In this review, we aim to compile the available information and reports focused on variations of the main gut microbiota taxa in metabolic diseases associated to xenobiotic exposure and related microbial metabolite profiles impacting the host health status. We performed an extensive literature search using SCOPUS, Web of Science, and PubMed databases and analysis of data. The data retrieval and thorough analyses highlight the need of more combined metagenomic and metabolomic studies revealing signatures for xenobiotics and triggered metabolic diseases. Moreover, metabolome and microbiome compositional taxa analyses allow further exploration of targeting beneficial NGP candidates according to their alleged variability abundance and potential therapeutic significance. Furthermore, this holistic review has identified limitations and needs of future directions to expand and integrate key knowledge to design appropriate clinical and interventional studies with NGP. Apart from human health, the identified beneficial microbes and metabolites could also be proposed for various applications under One Health, such as probiotics for animals, plants and environmental bioremediation.

Introduction: Main paragraph/s adressing the issue.

We think that this work is significant and necessary, but complex because it tries to integrate different key data available from topics independently studied. There are scarce data that show the link between the impact of xenobiotics in host health considering the role of the individualized microbiota taxa, pathways and key metabolites triggering diseases and susceptible to be modulated and become interventional biomarker. Therefore, the main aim of this work was to compile data, identify and describe the potential association between environmental and dietary xenobiotic exposure, gut microbiota taxa, and gut microbiota metabolites, taking into account host health implications and approaching novel biological strategies to restore gut microbiota dysbiosis and the induced dysfunctions.

  • I don't like the title and I believe it may confuse the reader

DONE. The title has been change for targeting the main analysis and key ideas.

  • Define "endocrine disorders". It's very broad.

Considering your suggestions and for avoiding possible confusions, we have been focused on metabolic diseases instead endocrine disorders, because the interest of this manuscript is the general impact of xenobiotics in microbiota modifications and related metabolic diseases.

Fort he sake of better understanding and simplifying the concepts, we have written along the text “metabolic disorders and diseases”, knowing that endocrine disorders are also promoting  metabolic diseases.

  • In addition, the authors mention the taxonomy ("taxa") of the Microbiome, while the most coherent term would be "Microbiota". It would be good to review the term "Microbiome taxa" throughout the text and make changes to the title.

IT HAS BEEN CORRECTED AS SUGGESTED AS WE ALSO CONSIDER THE TERM MORE APPROPRIATE. THANKS.

  • The discussion is portrayed in a superficial way, often even non-specific, with little grounded theoretical reference (references); it is possible to improve.
  • The "Next Generation Probiotics (NGP)" mentioned are little explored and this is an extremely important content that maybe could have an improved discussion.
  • The discussion is relatively short, it can be further explored.

THE DISCUSSION HAS BEEN ENLARGED ACCORDING TO THE MAIN SUGGESTIONS. THANKS SO MUCH.  

The authors reviewed the changes in the diversity of the intestinal microbiota from a taxonomic point of view, as well as the profile of metabolites produced, both under the influence of xenobiotics or under conditions of metabolic disorders. In addition, they bring the proposal of the use of probiotics as a therapeutic alternative.

  • However, I suggest including the subject "Future Perspectives" at the end of the manuscript, seeking to further explore what, which and how NGPs can be a promising alternative for many diseases (a fact that is unclear),

DONE. A section of Future Perspectives has been included

  • Also exploring the clinical application of this study.

DONE. The clinical trials and potential therapeutical applications has been explictly described along the manuscript and mainly addressed in the discussion and future perspectives. Moreover, table 2 showed specifically doses of NGP in order to highlight the importance of the administration parameters.

  • Do the authors consider that there are any limitations of the study? If yes, include it in the manuscript.

DONE. A section of Limitation of the study has been included

  • The conclusion is generic, tends towards a more superficial content, that is, non-specific.

DONE. THANKS FOR THE SUGGESTION. We have enlarged and modified the conclusions. We summarised better the main ideas extracted from the work done. The conclusions show now a better and comphrehensive meta-analysis.

Reviewer 3 Report

General comments:

The review aims to propose the association among xenobiotics intake, metabolic-endocrine disorders, and gut microbiota taxa. People who have endocrine disorders have dysbiosis in their gut. On the other hand, harmful xenobiotics, such as pesticides and fungicide cause endocrine disorders. Recent studies have suggested the intake of xenobiotics changes our gut microbiota. Thus, changes in gut microbiota may be associated with the cause of endocrine disorders. As a treatment of endocrine disorders, the next generation probiotics by neutralizing the dysbiosis is expected. The authors assessed with comprehensive literature reviews of gut microbiota studies and are suggesting the link among gut microbiota taxa, metabolic-endocrine disorders, and xenobiotics intake.

Specific comments:

To be frank, I had difficulty to understand what the authors would like to propose in this review. Do the authors suggest that people having endocrine disorder and people exposed by xenobiotics tend to have similar dysbiosis in their gut? Thus, would the authors like to conclude that gut microbiota is one of the main causes for endocrine disease? Due to the similarly (correct me if I am wrong, to me table 1 and 2 do not like similar), would the authors like to propose some gut bacterium can be the next generation probiotics to treat endocrine disease? I cannot get from this review what kind of association the authors are proposing. The reasoning of this proposal is not clear. Some chemicals cause endocrine disease, and patients with endocrine disease and people exposed by xenobiotics have similar dysbiosis in their gut. Thus, would the authors like to conclude that these three are associated? I also would like to suggest changing the title for readers.

L258-261 Conclusions “The interplay between metabolic and molecular pathways involving xenobiotics degradation, gut microbiota and metabolome is a target to be continuously studied to develop new treatments for metabolic and endocrine diseases” I think there are lots of pathways are involved in this topic. It would be nice if this review shows more clearly what pathways can be targeted for developing new treatments. Tables including the supplemental are too much description, thus difficult to follow what is important. For example, L130-151 “3.1.1. Microbiota taxa modifications in metabolic disorders… 3.1.2. Combined analysis of microbiota taxa and metabolome profiles in metabolic disorders and pathologies…”. L164-168 “3.1.3. Microbiome and metabolite profiles linked to xenobiotics exposure.” These are almost impossible for readers to follow everything. The authors should visualize these information. A graph like Figure 1 would be useful.

L262-266 Conclusions “The use of selected NGP, after knowing its functionality and molecular mechanisms, seems to be a promising strategy to restore homeostasis of the intestinal microbiota, taking into account food safety and risk assessment studies and their clinical impact in murine models and subsequently confirmations in human studies.” As a result of the comprehensive literature review, several candidates for the next generation probiotics are listed in Table 3. Especially, Akkermansia muciniphila is highlighted. The authors described in L250-252 “Considering each of the candidates individually, A. muciniphila is one of the better studied and most relevant potential microorganisms, due to the effects attributed to this microorganism, under certain conditions of dose and viability of the bacterial cells.” If the authors think that Akkermansia muciniphila is a promising target as the next generation probiotics, I would suggest making a paragraph to summarize the current studies of Akkermansia muciniphila. This bacterium has been considered as the next-generation beneficial gut bacterium. It would encourage the future studies if the authors highlight.

Table 3 is helpful to select the next generation probiotics. Other than Akkermansia muciniphila, it looks like “fatty acid lipid metabolites” modified by Faecalibacterium and Bacteroides are also a potential therapeutic target. However, the contribution of Faecalibacterium and Bacteroides seems different from Akkermansia muciniphila. Fatty acid lipid metabolites do not seem to be major modification in table 1 and 2, but in table 3 it is one of the main players. For the neutralization strategies for dysbiosis, how is targeting fatty acid lipid metabolites promising? 

Author Response

REVIEWER 3

AUTHORS REPLY

General comments: The review aims to propose the association among xenobiotics intake, metabolic-endocrine disorders, and gut microbiota taxa. People who have endocrine disorders have dysbiosis in their gut. On the other hand, harmful xenobiotics, such as pesticides and fungicide cause endocrine disorders. Recent studies have suggested the intake of xenobiotics changes our gut microbiota. Thus, changes in gut microbiota may be associated with the cause of endocrine disorders. As a treatment of endocrine disorders, the next generation probiotics by neutralizing the dysbiosis is expected. The authors assessed with comprehensive literature reviews of gut microbiota studies and are suggesting the link among gut microbiota taxa, metabolic-endocrine disorders, and xenobiotics intake.

THANKS SO MUCH FOR THE POSITIVE EVALUATION AND ALL SUGESTIONS TO IMROVE THE WORK DONE

  • Specific comments:To be frank, I had difficulty to understand what the authors would like to propose in this review. Do the authors suggest that people having endocrine disorder and people exposed by xenobiotics tend to have similar dysbiosis in their gut? Thus, would the authors like to conclude that gut microbiota is one of the main causes for endocrine disease?

WE HAVE REVISED THE WORK WITH THESE SUGGESTIONS AND TRYING TO FACILITATE THE READING AND THE USEFULNESS OF THE STUDY FOR THE SCIENTIFIC COMMUNITY.

THE HYPOTESIS PROPOSED IS THAT MICROBIOTA DYSBIOSIS OR ITS ALTERATION IS A PLAYER OR ACTOR THAT IS LINKED TO THE SEVERITY OF METABOLIC DISEASES CAUSED BY XENOBIOTICS. WHEN THE XENOBIOTIC IS A ENDOCRINE DISRUPTOR THE DISORDER TRIGGERED IS CONSIDERED AN ENDOCRINE DISEASE. HOWEVER, TO SIMPLIFY THE CONCEPTS WE HAVE KEEP ONLY METABOLIC DISEASES AS TARGET IN THE STUDY.

DOING AVAILABLE AND COMPILING MORE DATA FOR SUPPORTING THIS HYPOTHESIS AND BASED ON MICROBIOTA AND METABOLITES ANALYSIS WILL ALLOW TO BETTER FIND THE ASSOCIATION AND/OR CAUSALITY BY ESTABLISHING SIGNATURES.

THE MODIFICATIONS TO HIGHLIGHT THE SCOPE HAVE BEEN DONE IN THE ABSTRACT, INTRODUCTION, FIGURES, TABLES, DISCUSSION, CONCLUSIONS.

  • Due to the similarly (correct me if I am wrong, to me table 1 and 2 do not like similar), would the authors like to propose some gut bacterium can be the next generation probiotics to treat endocrine disease?

THANKS SO MUCH FOR THIS SUGGESTIONS. WE HAVE CORRECTED ACCORDINGLY

AS PREVIOUSLY STATED: THE STUDY IS IN EXPLORATORY PHASE, BUT ONLY ESTABLISHING MORE DATA FOR SUPPORTING THE HYPOTESIS WILL DRIVE SCIENTIFIC COMMUNITY TO PERFORM NEXT STEPS AND CLINICAL EXPERIMENTAL RESEARCH.  

* I cannot get from this review what kind of association the authors are proposing. The reasoning of this proposal is not clear. Some chemicals cause endocrine disease, and patients with endocrine disease and people exposed by xenobiotics have similar dysbiosis in their gut.

  • Thus, would the authors like to conclude that these three are associated?

THANK FOR POSING THE QUESTION.

YES, MICROBIOTA MEDIATES THE IMPACT OF THE XENOBIOTICS DIFFERENTIALLY ACCORDING TO THE INDIVIDUALIZED PROFILES.

THE MODIFICATIONS TO HIGHLIGHT THE RELEVANCE OF THIS IDEA HAVE BEEN DONE IN THE ABSTRACT, INTRODUCTION.

  • I also would like to suggest changing the title for readers. DONE

L258-261 Conclusions “The interplay between metabolic and molecular pathways involving xenobiotics degradation, gut microbiota and metabolome is a target to be continuously studied to develop new treatments for metabolic and endocrine diseases” I think there are lots of pathways are involved in this topic. It would be nice if this review shows more clearly what pathways can be targeted for developing new treatments.

A NEW FIGURE AND  SEVERAL PARAGRAPHS HAVE BEEN INCLUDED IN THE DIFFERENT SECTIONS TO STATE THE POTENTIAL INVOLVED PATHWAYS. 

Figure 3. Scheme of microbiota taxa and metabolites analysis with open access tools (KEGG, HGMA, BioCyc ). Main microbial metabolisms affected in metabolic diseases (lipid, bile acids, acetyl-CoA and amino acids metabolisms).

The analysis of these NGP studies seem to mitigate several metabolic disorders by restoring gut microbiota dysbiosis and modifying specific metabolisms. The altered microbiota taxa cause disturbances mainly on lipid metabolism by increasing substrates for energy metabolism in the liver and peripheral tissues. Data extraction shows that alteration on bile acid metabolism also affect the digestion of dietary lipids in the gut and other signaling functions. The gut microbiota has a deep effect on bile acid metabolism by promoting deconjugation, dehydrogenation, and dehydroxylation of primary bile acids. Other dysbiotic gut bacterial compositional profile can play an important role in susceptibility to metabolic disorders by affecting amino acid bioavailability to the host.

The SCFA metabolites produced by NGP seem to be key player on lipid metabolism as metabolic endpoints modulators [57]. However, it is also important to analyse the regulation of bile acid metabolism by NGP [58]. Acetyl-CoA derivatives [59] and amino acids bioavailability to the host, because specific microbiota taxa could also determine the susceptibility to linked metabolic disorders [60].

Relevant modifications of potential signature metabolites performed by targeted microbiota taxa belong to lipid, bile acid, acetyl-CoA and amino acid metabolisms.

  • Tables including the supplemental are too much description, thus difficult to follow what is important. For example,L130-151 “3.1.1. Microbiota taxa modifications in metabolic disorders…
  • 1.2. Combined analysis of microbiota taxa and metabolome profiles in metabolic disorders and pathologies…”.
  • L164-168 “3.1.3. Microbiome and metabolite profiles linked to xenobiotics exposure.” 
  • These are almost impossible for readers to follow everything. The authors should visualize these information. A graph like Figure 1 would be useful.

THANKS FOR THE SUGGESTIONS. WE HAVE ALL MODIFIED ACCORDINGLY  

WE HAVE DELETED TABLE S2 AND S3 FOR THE SHAKE OF SIMPLIFYING THE READING. DATA AVAILABLE IN TABLE 1 AND 2 IN THE MAIN BODY TEXT WILL BE ENOUGH TO A COMPREHENSIVE UNDERSTANDING OF KEY MESSAGES TOGETHER WITH EXPLAINING PARAGRAPHS ADDED.

  • L262-266 Conclusions “The use of selected NGP, after knowing its functionality and molecular mechanisms, seems to be a promising strategy to restore homeostasis of the intestinal microbiota, taking into account food safety and risk assessment studies and their clinical impact in murine models and subsequently confirmations in human studies.” 

As a result of the comprehensive literature review, several candidates for the next generation probiotics are listed in Table 3. Especially, Akkermansia muciniphila is highlighted. The authors described in L250-252 “Considering each of the candidates individually, A. muciniphila is one of the better studied and most relevant potential microorganisms, due to the effects attributed to this microorganism, under certain conditions of dose and viability of the bacterial cells.” 

If the authors think that Akkermansia muciniphila is a promising target as the next generation probiotics, I would suggest making a paragraph to summarize the current studies of Akkermansia muciniphila. This bacterium has been considered as the next-generation beneficial gut bacterium. It would encourage the future studies if the authors highlight.

THANKS FOR THESE VALUABLE SUGGESTIONS. WE HAVE INSERTED FULL PARAGRAPHS FOR HIGHLIGHTING THE RELEVANCE OF Akkermansia muciniphila (IN RESULTS, DISCUSSION).

  • Table 3 is helpful to select the next generation probiotics. Other than Akkermansia muciniphila, it looks like “fatty acid lipid metabolites” modified by Faecalibacterium and Bacteroides are also a potential therapeutic target.

However, the contribution of Faecalibacterium and Bacteroides seems different from Akkermansia muciniphila. Fatty acid lipid metabolites do not seem to be major modification in table 1 and 2, but in table 3 it is one of the main players.  For the neutralization strategies for dysbiosis, how is targeting fatty acid lipid metabolites promising? 

WE UNDERSTAND THESE QUESTIONS, THERE IS A SIMILAR TREND, BUT THERE IS NO YET ENOUGH STUDIES TO CONCLUDE ON THE SAME EFFECTS OR MODE OF ACTIONS FOR ALL NGP.

THIS IS ONE OF THE MAIN LIMITATIONS AND REASONING BEHIND THIS EXTENSIVE REVIEW TO HIGHLIGHT THE NEED TO COMBINE MICROBIOME AND METABOLOME STUDIES TO KNOW BETTER THE PHISIOLOGYCAL IMPACT.

WE HAVE INTRODUCED THESE ISSUES IN THE DISCUSSIONS AND MAIN LIMITATIONS.

Round 2

Reviewer 3 Report

Please see an attached PDF file as my comments. 

Author Response

AUTHORS: WE WOULD LIKE TO THANK YOU FOR YOUR POSITIVE EVALUATION OF THE REVISED MANUSCRIPT. 

REVIEWER COMMENTS TO BE ADDRESSED 

The new title is an overstatement. “Metabolic and microbiota dysbiosis by xenobiotic exposure can be modulated by Next Generation Probiotics” The authors are suggesting this message by literature review, but this is not conclusive yet due to lack of experimental evidence. Thus, this topic needs more studies for the future to demonstrate. I suggest the authors to make a title based on the facts. 

AUTHORS RESPONSE:

WE HAVE MODIFIED THE TITLE, IT IS NOW MORE REALISTIC ACCORDING TO THE CONTENT, AS SUGGESTED:  

Exploring Next Generation Probiotics for metabolic and microbiota dysbiosis linked to xenobiotic exposure: holistic approach

REVIEWER COMMENTS TO BE ADDRESSED 

  • The new figure 3 for scheme is impossible to read. I can understand main microbial metabolism part (bottom), but not the others. The authors should improve this figure for readers.

AUTHORS RESPONSE

NEW FIGURE HAS BEEN BUILT KEEPING THE KEY MESSAGE. 

* MOREOVER, ENGLISH CHECK EDITION HAS BEEN DONE